# Performance and Stability of Tenofovir Alafenamide Formulations within Subcutaneous Biodegradable Implants for HIV Pre-Exposure Prophylaxis (PrEP)

**DOI:** 10.3390/pharmaceutics12111057

**Published:** 2020-11-05

**Authors:** Linying Li, Leah M. Johnson, Sai Archana Krovi, Zach R. Demkovich, Ariane van der Straten

**Affiliations:** 1Engineered Systems, RTI International, 3040 E Cornwallis Road, Research Triangle Park, Durham, NC 27709, USA; ali@rti.org (L.L.); akrovi@rti.org (S.A.K.); 2Women’s Global Health Imperative, RTI International, 2150 Shattuck avenue, Berkeley, CA 94704, USA; zdemkovich@rti.org (Z.R.D.); ariane@rti.org (A.v.d.S.)

**Keywords:** HIV pre-exposure prophylaxis, tenofovir alafenamide, implant, long-acting drug delivery, poly(ε-caprolactone) (PCL), biodegradable polymer

## Abstract

A critical need exists to develop diverse biomedical strategies for the widespread use of HIV Pre-Exposure Prophylaxis (HIV PrEP). This manuscript describes a subcutaneous reservoir-style implant for long-acting delivery of tenofovir alafenamide (TAF) for HIV PrEP. We detail key parameters of the TAF formulation that affect implant performance, including TAF ionization form, the selection of excipient and the exposure to aqueous conditions. Both in-vitro studies and shelf stability tests demonstrate enhanced performance for TAF freebase (TAF_FB_) in this long-acting implant platform, as TAF_FB_ maintains higher chemical stability than the TAF hemifumarate salt (TAF_HF_). We also examined the hydrolytic degradation profiles of various formulations of TAF and identified inflection points for the onset of the accelerated drug hydrolysis within the implant using a two-line model. The compositions of unstable formulations are characterized by liquid chromatography-mass spectrometry (LC-MS) and are correlated to predominant products of the TAF hydrolytic pathways. The hydrolysis rate of TAF is affected by pH and water content in the implant microenvironment. We further demonstrate the ability to substantially delay the degradation of TAF by reducing the rates of drug release and thus lowering the water ingress rate. Using this approach, we achieved sustained release of TAF_FB_ formulations over 240 days and maintained > 93% TAF purity under simulated physiological conditions. The opportunities for optimization of TAF formulations in this biodegradable implant supports further advancement of strategies to address long-acting HIV PrEP.

## 1. Introduction

There are an estimated 38 million people presently living with HIV globally [1]. Although the annual numbers of newly infected people has steadily declined, 1.7 million new infections were reported in 2019 [2]. Progress for further reduction of new infections is challenging, as some countries have experienced rising rates and more than 50% of key populations still do not have access to current HIV prevention services [1,2]. This suggests that coverage with the currently available HIV prevention product regimens, such as oral pre-exposure prophylaxis (PrEP) is insufficient. A broader range of HIV prevention products that extend beyond daily oral therapy—such as long-acting (LA) injectables, implants and topical methods—could further reduce the rate of new infections overall and in key populations. These prevention products are at different stages of the development pipeline. The furthest along under regulatory review is the monthly dapivirine ring [3]. The bi-monthly cabotegravir injectable (CAB-LA) is in phase III trials and was shown recently to be more effective than daily oral PrEP [4]. Earlier in the LA-PrEP product pipeline are implants, which are typically rod-shaped drug delivery systems made from biocompatible material. Implants are inserted underneath the skin and reside subcutaneously to release therapeutic concentrations of drug over several months to years. Implants have been approved as contraceptives in women since 1983 in Finland and 1991 in the US [5,6]. They are now being developed for HIV PrEP by several research laboratories and organizations [6]. For example, Merck’s implant formulation of Islatravir, a highly potent nucleoside reverse transcriptase translocation inhibitor (NRTTI), delivered intracellular concentrations above the expected protective threshold in humans for 12 weeks and is estimated to last >1 year.

Efforts toward the development of tenofovir alafenamide (TAF) implants have been motivated, in part, by the successful implementation of the class of nucleotide reverse transcriptase inhibitor (NRTI) drugs for HIV treatment and PrEP. NRTI tenofovir (TFV) was the first antiretroviral compound used to demonstrate the efficacy of HIV PrEP in the macaque model [7,8]. TFV is phosphorylated to the active metabolite tenofovir diphosphate (TVF-DP) intracellularly, where it competes with 2’-deoxyadenosine triphosphate (dATP) and inhibits HIV-1 reverse transcriptase [9]. When administered orally, however, TFV had low bioavailability and poor cellular membrane permeability, which prompted subsequent efforts to develop prodrug forms of TFV to address these issues. Tenofovir disoproxil fumarate (TDF), the first generation prodrug of TFV, was approved for HIV treatment as Viread^®^ in 2001 and then for treatment and prevention in combination with emtricitabine as Truvada^®^ in 2012. The second generation prodrug, TAF, a phosphoramidate of TFV, was approved for HIV treatment as the fixed-dose combination Genvoya^®^ in 2015 and more recently for prevention as Descovy^®^ for at-risk adult and adolescent males. Compared to TDF, TAF is better suited for long-acting implants due to its superior potency [9,10,11,12], improved safety profile [13] and higher chemical stability at higher pH [14]. However, as a prodrug, TAF also possess the inherent hydrolytic instability of TDF and is particularly susceptible to acid hydrolysis [14]. Thus, retaining the stability of TAF under physiological condition over extended duration is essential for its application in long-acting delivery systems.

Several long-acting TAF implants are being developed for HIV PrEP, in the preclinical phase, where one is scheduled to begin Phase I/II human studies imminently [15,16,17,18,19,20]. In particular, Northwestern University described the safety, pharmacokinetics (PK) and manufacturing of a TAF_HF_ reservoir implant made of a permeable medical-grade polyurethane membrane with heat-sealed ends [16,20]. Oak Crest Institute of Science designed a non-permeable reservoir implant, made from medical grade-silicone, with delivery channels punched along the length of the device for TAF_FB_ to diffuse out [15]. This implant, called OCIS-001, is poised to be studied in a Phase I/II trial (PACTR201809520959443). The Houston Methodist Research Institute evaluated a nanochannel delivery implant (NDI) formed from a medical-grade titanium reservoir plugged with a silicone membrane housing a dense organized array of slit-nanochannels [17,21,22]. At RTI we are developing a biodegradable reservoir implant comprised of poly(ε-caprolactone) (PCL) for HIV PrEP. TAF has served as a model active pharmaceutical ingredient (API) for the platform and has demonstrated controlled release via tunning the surface area, thickness and molecular weight (MW) of the implant wall [18]. We have recently demonstrated zero-order delivery of TAF_FB_ over 6 months in-vitro and in vivo from biodegradable implants comprised of poly(ε-caprolactone) (PCL) [23].

Herein, we present advancements in the performance of a biodegradable implant for sustained delivery of TAF. We explore how the TAF ionization form (TAF_FB_ and TAF_HF_) along with the excipient type affects the performance of the implants, including in-vitro release rates and chemical stability of various TAF formulations. We demonstrate that TAF_FB_ outperforms TAF_HF_ within the reservoir-style implant by maintaining higher chemical stability under both simulated physiological and storage conditions. We assess the degradation profiles of TAF_FB_ and TAF_HF_ formulations and identified the predominant hydrolytic degradants of TAF. We attribute the degradation rate of TAF to be pH-dependent and inversely correlated to the rate of water ingress. Devices with thicker PCL walls are used to delay the hydrolysis of TAF within our implants. As a result, we did achieve a high level of TAF purity after 240-days of in-vitro exposure. This manuscript supports the continued advancement of new long-acting delivery systems to address HIV PrEP.

## 2. Materials and Methods

### 2.1. Solubility and Stability Screens

TAF freebase (TAF_FB_) and TAF hemifumarate salt (TAF_HF_) were graciously provided by Gilead Sciences (Foster City, CA, USA). Excipients Ethyl Oleate ES45252, Oleic Acid SR40211, Poloxamer 124 Synperonic PE/L 44 ETK1159, Castor Oil SR40890 (CO), Sesame Oil SR40294 (SO), Cottonseed Oil SR40166, PEG40 Castor Oil Etocas 40-SS-(MH)ET48333, PEG300 SR41329, PEG400 SR40377, PEG600 SR40269 (PEG600), Polysorbate 80SR48833, Propylene Glycol SR40836 were sourced from Croda (Snaith, UK) and Glycerol (catalog # G6279) was sourced from Sigma (St. Louis, MO, USA).

Excess TAF_FB_ (~40 mg) was mixed with neat excipient (~1 mL) in a 20 mL scintillation vial. Similarly, excess TAF_HF_ (~30 mg) was mixed with excipient (~0.4 mL). Each excipient mixture was incubated at 37 °C over a period of 2 days, after which the concentration of TAF_FB_ or TAF_HF_ was measured by high-performance liquid chromatography (HPLC) to determine the solubility of the API. To determine the stability of the API, the excipient mixtures were prepared as specified above and incubated at 37 °C for an additional 7 days prior to being analyzed by HPLC.

### 2.2. Implant Fabrication

Research-grade PCL pellets were purchased from Sigma Aldrich, referred to as “Sigma-PCL” throughout this paper (weight average molecular weight (*M*_w_) = 103 kDa, Cat# 440744, St. Louis, MO, USA) and in medical-grade from Corbion, referred to as “PC17” throughout this paper (average *M*_n_ = 93 kDa, PURASORB PC 17, Amsterdam, The Netherlands). PCL tubes were fabricated via a hot-melt, single screw extrusion process using solid PCL pellets at GenX Medical (Chattanooga, TN, USA). All tubes were 2.5 mm in outer diameter (OD) and had wall thicknesses of 70, 100, 150, 200 or 300 µm, as measured with a 3-axis laser measurement system and light microscopy at GenX Medical.

PCL tubes were sealed at both ends using injection sealing wherein the PCL tube was marked and trimmed to the correct length to achieve an implant with a 40 mm paste length with 3 mm of headspace at both ends for sealing. The initial seal was then created on one end of the implant by placing the tube over a stainless steel rod that filled all the tube except for a 3 mm headspace at one end, placing a Teflon collar around the headspace to support the tube wall and injecting molten PCL into the cavity of the headspace. After the injected PCL was solidified, excess PCL was trimmed and the collar was removed to form a cylindrical seal approximately 2 mm long that is compatible with commercial contraceptive trocars.

TAF_FB_ and TAF_HF_ were mixed with excipients at varying mass ratios prior to loading into the implant. Each mixture was first ground with a mortar and pestle to create a smooth paste and then backloaded into a 1 mL syringe fitted with a 14-gauge blunt tip needle. The TAF formulation was then extruded through the needle into the empty tube. Alternatively, the TAF formulation was loaded into the PCL tube using a modified spatula. After the filled formulation reached the 40-mm mark, the interior tube wall was cleaned with a rod and sealed in a similar manner to the first seal. After fabrication, all implants were weighed to determine the total payload and photographed with a ruler to record the final dimensions. Paste area was measured with ImageJ (Version 1.50e, National Institutes of Health (NIH), Bethesda, MD, USA) and release rates were normalized to the surface area of a full-sized implant (2.5 mm OD, 40 mm in length), 314 mm^2^. The end of the implants (i.e., end-seals) were not included in calculations of the implant surface area.

### 2.3. Implant Sterilization

All implants were fabricated and handled under aseptic conditions using a biosafety cabinet. Certain implants were exposed to gamma irradiation, as indicated in the text. Implants exposed to gamma irradiation were first packed in amber glass vials and then irradiated with a dose range of 18–24 kGy at room temperature, using a Cobalt-60 gamma-ray source (Nordion Inc., Ottawa, ON, Canada) at Steris (Mentor, OH, USA). Samples were exposed to the source on a continuous path for a period of 8 h.

### 2.4. In Vitro Release Studies

In vitro release characterization involved incubation of the implants in 40 mL 1X phosphate buffered saline (PBS) (pH 7.4) at 37 °C and placed on an orbital shaker. TAF species in the release media was measured by ultraviolet-visible (UV) spectroscopy at 260 nm using the Synergy MX multi-mode plate reader (BioTek Instruments, Inc, Winooski, VT, USA). The release buffer was sampled three times per week during which the implants were transferred to 40 mL of fresh buffer to maintain sink conditions. TAF quantity released in each PBS buffer during the time interval was calculated and cumulative mass of drug release as a function of time was determined. All the in-vitro release and stability studies have sample sizes of 20 implants, unless noted otherwise. At defined timepoints, 2 devices were taken down to determine chromatographic purity of TAF and water content inside the implant reservoir.

### 2.5. Stability Analysis of TAF Formulation

The purity of TAF formulations inside the implant reservoir was evaluated by slicing open an implant, extracting the entire reservoir contents into an organic solution and measuring TAF chromatographic purity using high performance liquid chromatography coupled with UV spectroscopy (HPLC/UV). The analysis was performed using a Waters BEH C18 column (2.1 mm × 50 mm, 1.7 μm) under gradient, reversed phase conditions with detection at 260 nm. Shelf stability implants containing TAF_FB_ formulations were analyzed using an Agilent Zorbax column (4.6 mm × 150 mm, 3.5 µm). For each implant, one single aliquot was prepared and quantitated by linear regression analysis against a five-point calibration curve. TAF purity was calculated as % peak area associated with TAF relative to total peak area of TAF related degradation products (detected above the limit of detection (LOD) ≥ 0.05%). The TAF formulations within the implant were analyzed after exposure of the implant to a simulated physiological condition (i.e., 1X PBS, pH 7.4 at 37 °C) for up to 240 days. A scalpel was used to slit the implants lengthwise and the contents of the scalpel were blotted on pH paper to assess the pH within the device core.

### 2.6. Loss on Drying Analysis

A glass-stoppered, shallow weighing bottle was placed in the vacuum oven at 40 °C for 30 min, cooled to room temperature and weighed (*W_bottle_*). Retrieved implants from each timepoint were placed in individual bottles and their weight was recorded (*W_w_*) and placed in the vacuum oven at 40 °C overnight (the glass stopper was removed from the bottle but left in the vacuum oven as well). The glass bottle was closed prior to weighing it with the implant (*W_d_*). This process was repeated until the new recorded weight (*W*’*_d_*) was within 0.1 mg of *W_d_*. The implant loss on drying (i.e., the amount of water content that was present in the implant) was calculated using the following equation:$$\%\;Loss\;on\;drying=\frac{W_{w}-W^{\prime }{}_{d}}{W_{w}-W_{bottle}}\times 100$$

### 2.7. Shelf Stability

Implant filled with TAF_FB_ and TAF_HF_ formulations were placed in aluminum foil pouches. Half of the pouches were sealed using an impulse heat sealer (AIE-110T, American International Electric Inc., Industry, CA, USA), while the remaining ones were left unsealed. The sealed pouches containing the implants were further split into two groups: stored under ambient conditions and in an incubator at 37 °C with 40% relative humidity (RH). The same was repeated for the unsealed pouches containing the TAF formulation implants. The implants were removed from the pouches and assessed for purity using HPLC at the following timepoints: 0, 90, 180 days. Note: the implants containing TAF_HF_ formulations were sterilized at > 40 kGy using the same parameters as specified in Section 2.3.

### 2.8. Differential Scanning Calorimetry (DSC)

The melting behavior of PCL samples was assessed with modulated differential scanning calorimetry (MDSC) (TA Instruments Q200, RCS90 cooling system, New Castle, DE, USA). Approximately 8 mg of extruded polymer tubing was placed in a Tzero™ Pan and sealed with a Tzero™ Lid and a dome-shaped die, resulting in a crimped seal. Samples were then placed in a nitrogen-purged DSC cell, cooled to 0 °C, then heated to 120 °C at a rate of 1 °C/min with an underlying heat-only modulation temperature scan of ±0.13 °C every 60 s. The melting temperature (*T_m_*) of the polymer was determined by the peak temperature of the melting endotherm and the enthalpy associated with melting was determined by integrating linearly the area of the melt peak (between 25 and 65 °C) using the TA Universal Analysis software (version 4.5A, TA Instruments, New Castle, DE, USA). PCL samples did not exhibit exothermic peaks in the non-reversing heat flow signal indicating that PCL did not experience cold-crystallization during the melting process; therefore, the total heat flow curve was used to assess the mass % crystallinity. The mass % crystallinity was calculated using the following equation, where *X_c_* represents the mass fraction of crystalline domains in PCL, Δ*H_m_* represents the enthalpy of melting measured by the DSC and x*H_fus_* represents the theoretical enthalpy of melting for 100% crystalline PCL, reported as 139.5 J/g [24,25].
$$X_{C}=\frac{\Delta H_{m}}{\Delta H_{fus}}\times 100.$$

### 2.9. Gel Permeation Chromatography (GPC)

The MW of PCL was analyzed via GPC by first dissolving samples in tetrahydrofuran (THF) to 10 mg/mL injecting 40 µL of sample using an Agilent 1100/1200 HPLC-UV instrument (Santa Clara, CA, USA, flow rate of 1.0 mL/min). Polystyrene polymer standards (498 Da to 554 kDa) were used to calibrate the MW of samples.

## 3. Results and Discussion

### 3.1. In Vitro Performance of PCL Reservoir Implants with TAF_HF_ and TAF_FB_ Formulations

In this study, we investigated various TAF formulations within a biodegradable implant under simulated physiological conditions. The implant configuration comprises a reservoir of formulated API encapsulated by a biodegradable PCL membrane (Figure 1). The implants were fabricated from PCL tubes produced via a hot-melt, single screw extrusion process using solid PCL pellets, with an outer diameter (OD) of 2.5 mm and a length of 40 mm. Implants containing TAF in solid form with no additional excipients were first evaluated under in-vitro conditions that mimic physiological environments. Implants containing only TAF exhibited non-linear release profiles (see Appendix A), likely a consequence of the dissolution process of solid TAF within the reservoir and the lack of PCL membrane-controlled release of drug. Therefore an excipient was incorporated into the API formulation to tailor dissolution rates. We conducted a screening study with several excipients identified from the FDA’s inactive ingredient list. The excipient screen involved mixing TAF_FB_ or TAF_HF_ with various excipients and incubating the mixtures at 37 °C. The solubility and stability of each TAF form within the pharmaceutical grade excipients were determined by a HPLC method after 2 or 9 days of incubation, respectively (Table 1). Lead excipients were identified for each API with criteria of showing < 3% impurity level. Sesame oil and castor oil were therefore down-selected for TAF_FB_. Besides the castor oil, a poly(ethylene glycol) (PEG) excipient was also considered for TAF_HF_, because of a high TAF_HF_ stability reported by Schlesinger et al., within the thin-film polymer implants [26]. We selected PEG600 as an excipient for TAF_HF_, due to its larger molecular weight as compared to other PEG excipients of shorter chain length (i.e., PEG300, PEG400).

After selection of the excipients, we conducted an in-vitro study to assess the release kinetics and the chromatographic purity of TAF_HF_ and TAF_FB_ within the reservoir over time when implants were immersed in aqueous, physiologically relevant conditions (pH = 7.4, 37 °C). TAF_HF_ and TAF_FB_ were first formulated with the lead excipients at a mass ratio of 2:1 and loaded in the PCL extruded tubes (100 µm wall thickness) comprising PCL with *M*_w_ of 145 kDa. Table 2 shows the formulation, TAF payload and configuration of tested implants. The cumulative release profiles for various TAF formulations are shown in Figure 2. All implants exhibited a period of zero-order drug release. TAF_FB_ implants exhibited linear release with a constant release rate of TAF species (non-degraded TAF and tenofovir-containing species) over 210 days, while the TAF_HF_ implants demonstrated zero-order release of TAF species up to 120 days. The drug release rates for implant containing formulations with TAF_HF_ and PEG600 exhibited relatively large variations and deviated from zero-order release near day 90, which was likely attributed to the swelling of the implants. This swelling behavior was previously observed for other hydrophilic excipients (i.e., glycerol, PEG300, PEG400) due to the high-water solubility, which leads to high osmotic pressure within the implant core. The average release rates for TAF_HF_-CO formulation and TAF_HF_-PEG600 formulation were 0.38 ± 0.04 mg/day and 0.68 ± 0.20 mg/day, respectively. TAF_FB_ formulations exhibited lower release rates than the TAF_HF_ formulations, with average release rates of 0.26 ± 0.04 mg/day and 0.18 ± 0.03 mg/day for castor oil and sesame oil formulation, respectively. The differences in release rates between implants containing TAF_HF_ and TAF_FB_ are likely related to the solubility of these APIs within the excipients and PBS buffer, since TAF_HF_ showed a higher solubility within PBS (11.6 mg/mL) than TAF_FB_ (5.8 mg/mL). In addition, the TAF_FB_-CO formulation demonstrated a faster release rate than the TAF_FB_-SO implants, which is also likely attributed to the higher solubility of the TAF_FB_ within castor oil. According to Fick’s first law of diffusion, the release rate is directly proportional to the drug concentration gradient across the PCL membrane, which is equivalent to the solubility of API within the excipients when zero-order release kinetics is achieved [27]. As shown in Table 1, TAF_FB_ showed higher solubility within castor oil than sesame oil, resulting in higher release rates. Similarly, the TAF_HF_-PEG600 formulation demonstrated a faster release rate than the TAF_HF_-CO implants, which is likely attributed to the higher solubility of the TAF_HF_ within PEG600.Therefore, the release rate of the implant is dictated by the solubility of the API within the selected excipients and the release media. The excipient choice is critical for tuning the release rate of TAF.

### 3.2. In Vitro Stability Assessment of Implant Formulations

To evaluate the degradation profiles of TAF_HF_ and TAF_FB_ inside the reservoir of implants exposed to PBS at 37 °C, implants were periodically removed from the study and sacrificed to assess the stability of TAF using HPLC analysis. Figure 3a,b shows the chromatographic purity of TAF_HF_ in the reservoir of implants over 120 days. By day 120, the purity of TAF_HF_ decreased from 99.8% to 17.6% for TAF_HF_-CO formulations and from 99.7% to 1.36% for TAF_HF_-PEG600 formulations (raw data shown in Appendix A). These results show that TAF_HF_-CO formulations exhibit a higher degree of stability than the TAF_HF_-PEG600 formulations after exposure to simulated physiological conditions, which is consistent with the stability results from the excipient screen. It is reasonable to expect that TAF_HF_-PEG600 formulation degrades at a faster rate given the hydrophilicity of PEG600. In comparison, the chromatographic purity of TAF_FB_ formulations was also monitored using the HPLC method. Results are shown in Figure 3c,d and raw data are listed in Appendix A. Unlike TAF_HF_ formulations, a high level of purity was achieved for TAF_FB_ formulations at 210 days. The TAF_FB_-SO formulation demonstrated a slightly higher purity (92%) than that of TAF_FB_-CO formulations (85%) at 210 days, which is in a good agreement with the excipient screen results.

Irrespective of the formulation, TAF within the implants remained unhydrolyzed and chemically stable within the implant at the beginning of the in-vitro study, then reached a point in time when the degradation rate accelerated. These results are consistent with our hypothesis of a two-stage degradation profile of TAF, where TAF in the solid-state remains relatively stable and then hydrolyzes to TAF degradants once solid TAF dissolves in aqueous solutions. To assess the degradation rate constants for the initial and accelerated stages, the chromatographic purity of TAF was plotted as a function of time and a two-line model was applied to depict the non-linear degradation profiles of various TAF formulations. Excellent fits to the experimental data were obtained (R^2^ > 0.94 for all the fittings, see Figure 3). The two-line model also identified the inflection point of these degradation profiles, which represents the point in time when accelerated degradation of TAF begins.

As expected, TAF_HF_-PEG600 formulation exhibited a higher initial rate of degradation with an early inflection point at ~38 days, compared to ~78 days for TAF_HF_-CO formulation. Furthermore, the two-line models can project the purity of TAF_HF_ along the time axis for these specific formulations. It is predicted that TAF_HF_ would be completely hydrolyzed and depleted within the 100 μm thick implants with the TAF_HF_-CO formulation after exposure to the in-vitro environment for ~140 days. In contrast, TAF_FB_ formulations exhibited a lower rate of degradation with a delayed onset of the accelerated degradation phase (~130 days). Based on the model, the TAF_FB_-SO and TAF_FB_-CO formulations can maintain > 90% purity for up to 220 days and 155 days, respectively. In particular, the TAF_FB_-SO formulation showed a lower rate of degradation for both initial and accelerated phases of degradation, as compared to TAF_FB_-CO formulation. These results indicate that TAF_FB_ formulations are more stable than the TAF_HF_ formulations under simulated physiological conditions. This is likely attributed to the differences in the hydrolytic instability of TAF_FB_ and TAF_HF_ formulations within the device cores, as both TAF_FB_ and TAF_HF_ degrade through hydrolysis [9,14]. Specifically, hydrolytic degradation and pathway of TAF are dependent on the pH in the implant microenvironment [22]. The degradation rate of TAF increases in basic conditions due to the P-O bond in its structure that is prone to hydrolysis in alkaline conditions [28]. In addition, a higher instability of TAF in low-pH conditions is also observed and is likely due to the presence of P-N (phosphoramidate) bond, which particularly is susceptible to acid hydrolysis [29]. Thus, a pH “stability window” for TAF between pH 4.8–5.8 was determined by Grattoni et al. [22]. To elucidate why TAF_FB_ formulations outperformed TAF_HF_ formulations within our platform, we further measured the pH of the formulated implant core using pH paper prior to purity analysis. Both TAF_FB_ formulations showed similar pH levels of ~4–5 at 210 days, whereas the pH of residual TAF_HF_ formulation inside the device core was ~2–3 at ~200 days (data not shown). Implant reservoir loaded with TAF_HF_ formulation is expected to have a more acidic environment due to fumaric acid. This explains the low stability of TAF_HF_ formulation because lower pH leads to more rapid TAF degradation.

Besides the pH level of the microenvironment in the device core, we also hypothesized that a correlation exists between the level of TAF degradation and the amount of water uptake by the implants. Therefore, we measured the water content within the implants using a loss-on-drying method. This approach is adapted from the United States Pharmacopeia (USP) Chapter for Loss-on-Drying, where we compared the weight of an implant before and after it was dried in a vacuum oven at 40°C after removal from buffer. The percentage of water content for various TAF formulations was plotted as a function of time (Figure 4). The amount of water ingress at different time points along with the cumulative amounts of drug release at a given time is also listed in Appendix A. At Day-120, the TAF_HF_-CO implants gained ~10 mg of water, whereas TAF_HF_-PEG600 implants gained ~18 mg of water. The significantly large amount of water uptake by the TAF_HF_-PEG600 implants explains the low levels of chromatographic purity of TAF_HF_ inside the implant reservoir. In contrast, TAF_FB_ implants measured a much lower percentage of water ingress as compared to TAF_HF_ implants at a given time. For instance, the amount of water ingress at 120 days for TAF_FB_-CO and TAF_FB_-SO only consists of ~4.6% and 8.4% of the total mass of the implants, respectively, which are significantly lower than that of TAF_HF_-CO (~11.6%) and TAF_HF_-PEG600 implants (~21.6%). In addition, TAF_FB_ -SO formulation exhibited a lower level of water ingress than the TAF_FB_-CO formulation, which is likely due to a lower release rate of the TAF_FB_-SO formulation. As shown in Appendix A, the amount of cumulative drug release for TAF_FB_-CO formulation is higher than TAF_FB_-SO formulation at a given time. Although both TAF_FB_-SO and TAF_FB_-CO formulation exhibited a comparable pH level in the implant core (pH of ~4–5), higher drug release may create more void space for water to permeate into the implants resulting in faster degradation of TAF_FB_. Thus, reducing the release rate of drug from the implants can potentially slow down the rate of water ingress and improve the formulation stability.

Similar to the TAF degradation profiles, the water ingress profiles also showed a two-step process, where the amount of water ingress was negligible at the beginning of the study, then increased dramatically at an accelerated rate (Figure 4). To assess the water permeation rates, two-line models were also applied to the water ingress profiles of various formulations. Excellent fits to the water ingress data were obtained for the TAF_HF_-CO, TAF_FB_-CO and TAF_FB_-SO formulations (R^2^ > 0.92 for all the fitting). Table 3 shows the inflection point for each formulation along with the quantities of drug remaining, TAF purities and water content at the given inflection point. The inflection points in the water ingress profiles closely aligned with those of the degradation profiles, confirming that the TAF purity is correlated with the amount of water ingress. In contrast, the water ingress profile for the TAF_HF_-PEG600 implants is sigmoidal and does not fit the two-line models. As a hydrophilic and water-soluble excipient, PEG draws more water into the implant core, resulting in swelling of the implants. Additionally, four TAF_HF_-PEG600 implants that were subjected to the loss-on-drying analysis were compromised due to swelling, resulting in an artificially higher amount of water ingress. It is worth noting that TAF_FB_ formulations at the inflection point (~130 days) showed higher purity than the TAF_HF_ formulations at their inflection points (~38–78 days) while the percentages of water ingress are comparable for these formulations, indicating that TAF_FB_ is less susceptible to hydrolysis as compared to TAF_HF_. This is likely due to the differences in the pH level within the device core as previously discussed.

We further explored the relationship between the purity of TAF and the water content within these implants. Figure 5 illustrates the purity of TAF and the water content as a function of time for implants containing formulations of TAF_HF_ and TAF_FB_. Both the TAF degradation and water ingress profiles appear to exhibit a two-stage process. For example, the purity of formulated TAF_HF_ inside the implant reservoir decreases at an accelerated rate after ~80 days in-vitro, while the rate of water ingress also increases after the same time (see Figure 5a). Similar degradation behaviors were observed for TAF_FB_ around ~130 days for both castor oil and sesame oil formulations (Figure 5b). This data suggests that the predominant reason for accelerated TAF degradation is related to water ingress, wherein the accumulation of water within the reservoir accelerates the rate of TAF hydrolysis. In addition, the increasing concentration of TAF degradants within the implant core could in turn draw a larger amount of water into the implant, as the most prominent TAF degradants (e.g., tenofovir, monophenyl-TFV) are hydrophilic and/or water-soluble. A possibility also exists that the increased concentration of solute resulting from the breakdown of TAF creates an osmotic gradient as a driving force for water to imbibe into the implant core. Taken together, the degradation of TAF is caused by water ingress and the resultant degradation products may accelerate the rate of water permeation. Thus, a strong correlation between TAF impurity and water content within the implants was established. To summarize, as the degradation of TAF is dependent on the pH and amount of water ingress, implants containing the TAF_FB_-SO formulations showed a pH level within the “stability window” that substantially mitigates the TAF degradation and a relatively low release rate that slows down the rate of water ingress. Therefore, TAF_FB_-SO formulations were identified as the lead formulation for the sustained delivery of TAF within our biodegradable drug delivery platform.

### 3.3. Hydrolytic Degradants of TAF

In addition to assessing the chromatographic purity of TAF, individual TAF degradants > 0.05% were identified using known markers on HPLC with confirmation by liquid-chromatography mass spectrometry (LC-MS) at the relative retention time (RRT). Figure 6 shows the levels of individual degradants as a function of time for various TAF formulations. All degradants detected in the implant resulted from the hydrolysis of TAF, with the predominant species including TFV (the parent API) and monophenyl-TFV (an intermediate in the TAF hydrolytic pathway) for all formulations. These two degradants were measured at comparable levels within the TAF_FB_-CO, TAF_FB_-SO and TAF_HF_-CO implants, whereas the TAF_HF_-PEG600 implants contained a significantly higher level of monophenyl-TFV than TFV. The higher amount of monophenyl-TFV within the TAF_HF_-PEG600 implant is likely related to a larger amount of water ingress resulting from the compromised integrity of the implant. The observed TAF degradants are well-aligned with the degradation products in acid solution state reported in the literature. Figure 7 shows the postulated predominant TAF degradation pathway proposed by Golla et al. [14]. First, the phosphoramidate moiety of TAF undergoes hydrolysis to form monophenyl-TFV (RRT of ~0.6) on the release of alanine isopropyl ester (RRT of ~0.5), then monophenyl-TFV further undergoes phosphorus phenyl ester hydrolysis to yield TFV (RRT of ~0.36 and 0.05).

### 3.4. Six-Month Shelf-Stability of Implants with TAF Formulations

To assess the stability of TAF formulations under different storage conditions we conducted a 6-month shelf stability study of the lead TAF_HF_ and TAF_FB_ formulations identified in the in-vitro evaluations. All the TAF implants comprised Sigma-PCL with a wall thickness of 70 µm and formulations of a 2:1 weight ratio of TAF_HF_ to castor oil or a 2:1 weight ratio of TAF_FB_ to sesame oil. Implants were placed in open and closed foil pouches and then stored at 22 °C/50% RH and 40 °C/75% RH for six months. The long-term and accelerated storage conditions are selected based on FDA Guidance [30]. The purity of TAF within the implants was measured at 0, 90 and 180 days using the UPLC method. The chromatographic purity of each TAF formulation as a function of time is presented in Figure 8 and Appendix A. As expected, formulations for both the TAF_HF_ and the TAF_FB_ formulations demonstrated higher stability under storage conditions as compared to aqueous in-vitro conditions. When the package remained intact, the stability of TAF remained > 97% for all formulations. Conversely, implants in the opened pouches, representing an unintentional breach in the packaging, resulted in substantially different stability profiles between TAF_HF_ and TAF_FB_. At 180 days in accelerated stability conditions (40 °C/75% RH), the purity of TAF_HF_-CO substantially decreased to 91.2%, whereas the purity of TAF_FB_-SO remained at 97.5%. This result further shows that TAF_FB_ outperforms TAF_HF_ within the reservoir-style implant but also illustrates the importance of packaging design for future product translation efforts.

This shelf-stability study also identified the individual TAF degradants > 0.05% by RRT and LC-MS and showed the predominant degradants were monophenyl-TFV and TFV, as in the in-vitro studies. Figure 9 illustrates the levels of individual degradants at 180 days for various TAF formulations. Interestingly, TFV was detected at significantly higher levels than monophenyl-TFV for both TAF_FB_ and TAF_HF_ formulations under solid-state conditions of the shelf stability tests. For instance, TAF_FB_-SO formulations contained 1.31 ± 0.04% of TFV and 0.03 ± 0.002% of monophenyl-TFV at 180 days, while TAF_HF_-CO formulations showed 1.87 ± 0.77% of TFV and 0.85 ± 0.57% of monophenyl-TFV at 180 days.

### 3.5. Improving the Stability of TAF_FB_ Formulations

As discussed above, the degradation rate of TAF is pH-dependent and is related to the rate of water ingress. To further enhance the stability of TAF_FB_, Grattoni et al., included a trans-urocanic acid additive within an implant to preserve the optimal pH and maintain TAF purity > 90% in vitro for over 9 months [22]. We are currently exploring various pH modifiers and hydrophile lipophile balance (HLB) modifiers to further enhance the chemical stability of the TAF_FB_ formulations [31,32]. Here, to improve the stability of TAF, we evaluated the ability to mitigate the amount of the water ingress by reducing the release rate of TAF. Previously we have demonstrated that the release rates of TAF from the implant are inversely proportional to the wall thickness of the PCL tubes [18]. In this study, we evaluated the release rate of TAF_FB_-CO and TAF_FB_-SO formulations within PCL tubes at 150, 200 and 300 µm wall thickness. We used the PCL tubes comprising PC17, a medical-grade PCL with *M*_w_ of 93 kDa, to support future preclinical studies. Figure 10 shows the cumulative release profiles of the TAF_FB_-CO and TAF_FB_-SO implants at different wall thicknesses. Similarly, implants containing TAF_FB_-CO formulations exhibited higher release rates than TAF_FB_-SO implants. We also observed the inverse relationship between the thickness of PCL walls and the release rates of TAF for these medical-grade implants. For instance, as the wall thickness increased from 150 to 300 µm for implants comprised of TAF_FB_-CO formulation, the release rates of TAF decreased from 0.31 ± 0.06 mg/day to 0.10 ± 0.02 mg/day. The chromatographic purities of TAF_FB_-SO and TAF_FB_-CO formulations were assessed using the HPLC method at 210 and 240 days. As presented in Table 4, the chromatographic purity of TAF formulations is inversely correlated with the release rates of TAF, demonstrating the ability to delay the degradation of TAF by lowering the release rates of the implants and thus reducing the water ingress rate. After 240-day of in-vitro exposure, we achieved a purity of 93.2% for TAF_FB_-SO formulation within 300 µm implants. Importantly, thicker-wall implants also offer high mechanical strength and good device integrity. Although the release rate of 300 µm TAF_FB_-SO implants is relatively low, the therapeutic level of TAF could be potentially achieved using multiple implants that are subcutaneously inserted, similar to Probuphine^®^ [33] and Norplant^®^ [34]. Although the use of multiple, low-dose implants will not circumvent the unavoidable hydrolysis of TAF within these current implant formulations, the use of multiple implants to achieve the desired dosing could extend the therapeutic duration to longer periods of time with protection. It is worth noting the degradation profile of TAF formulations assessed under in-vitro conditions may not be reflective of the in-vivo conditions and efforts are currently underway to evaluate the degradation profiles of the these TAF formulations in preclinical studies using animal models (i.e., rabbit, dog, non-human primate) [23].

## 4. Conclusions

This manuscript highlights the importance of the ionization form of a drug when developing implantable drug delivery systems. Although the ionized hemifumarate salt form of TAF is currently used in clinically available oral formulations, we show that the non-ionized free base form of TAF is better suited for our reservoir-style implant due to higher chemical stability over time. The development of implants as LA drug delivery systems require API formulations that maintain a high degree of purity when exposed to physiological conditions over extended periods, in many cases for months to years. These studies show that the TAF_FB_ outperforms the TAF_HF_ by maintaining higher purity for a longer time in a reservoir-style implant. The reasons for the higher purity of TAF_FB_ is likely a result of multiple effects: slowed ingress rate of water due to lower release rates of TAF_FB_, the absence of the fumarate salt and achievement of an optimum pH window. To further delay the hydrolysis of TAF within our implants, we used thicker PCL walls to reduce the rates of drug release and water ingress. Using this approach, a purity of 93.2% of TAF was achieved with 300 µm implants comprising TAF_FB_-SO formulation (release rate of 0.07 mg/day) after 240-days of in-vitro exposure. In general, the delivery of hydrolysable drugs from an implant is feasible but requires careful consideration of attributes that could affect drug breakdown, including the implant form factor, mechanism of drug release, drug ionization form and environmental exposures (e.g., pH, temperature, water content). For the reservoir-style TAF implant presented in this paper, many of these parameters can be controlled to improve the stability of TAF and ultimately improve the performance of the implant for achieving long-acting HIV PrEP.

## Figures and Tables

**Figure 1 pharmaceutics-12-01057-f001:**
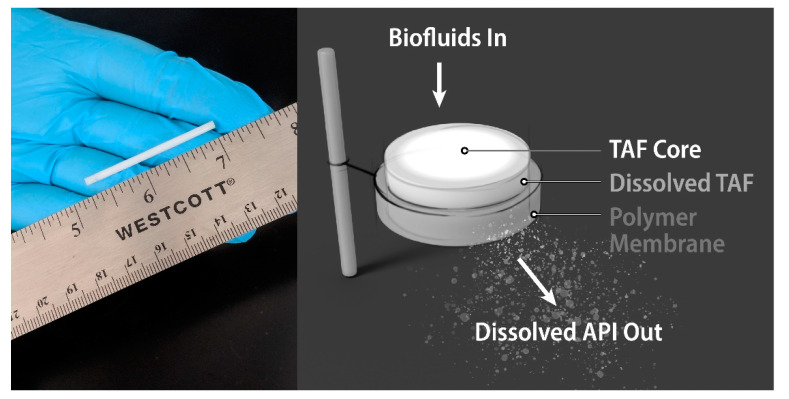
(**Left**) A digital camera image of the biodegradable implant. (**Right**) A schematic of a poly(ε-caprolactone) (PCL) reservoir-style implant for sustained delivery of tenofovir alafenamide (TAF) formulations.

**Figure 2 pharmaceutics-12-01057-f002:**
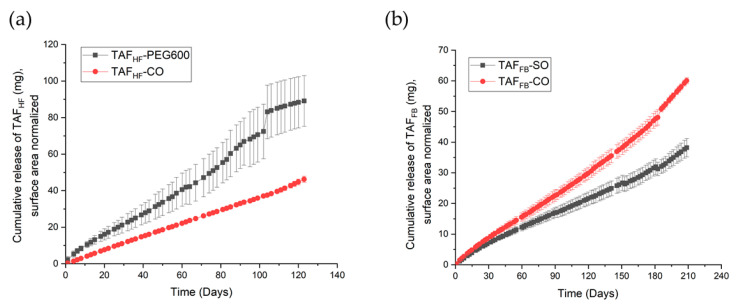
Cumulative release profiles of various TAF_HF_ (**a**) and TAF_FB_ (**b**) formulations.

**Figure 3 pharmaceutics-12-01057-f003:**
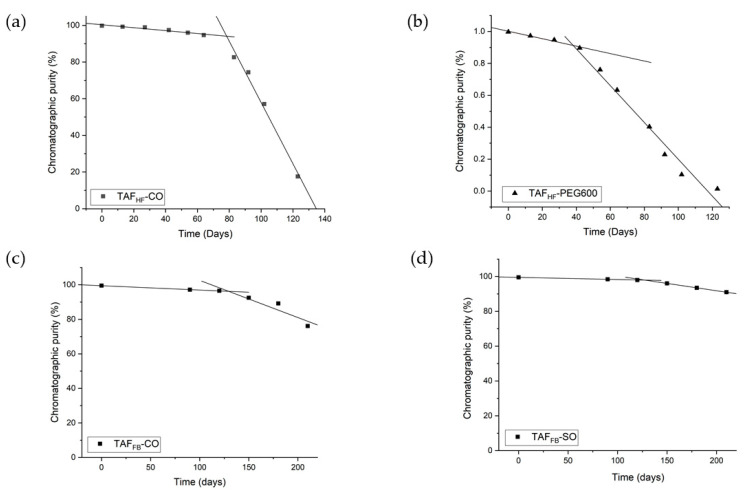
Degradation profiles of TAF_HF_ (**a**,**b**) and TAF_FB_ (**c**,**d**) formulations within implants with a wall thickness of 100 μm after exposure to simulated physiological conditions (37 °C, PBS). The fits (black lines) correspond to a two-line model for each formulation.

**Figure 4 pharmaceutics-12-01057-f004:**
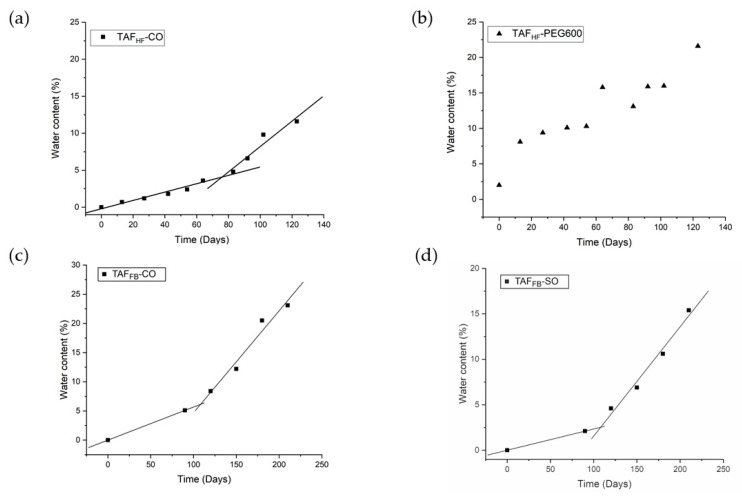
Water ingress profiles of TAF_HF_-CO (**a**), TAF_HF_-PEG600 (**b**), TAF_FB_-CO (**c**), and TAF_FB_-SO (**d**) formulations within PCL implants exposed to simulated physiological conditions (37 °C, PBS). Two-line model (depicted in black lines) has been applied to TAF_HF_-CO, TAF_FB_-CO and TAF_FB_-SO formulations in mass ratios of 2:1.

**Figure 5 pharmaceutics-12-01057-f005:**
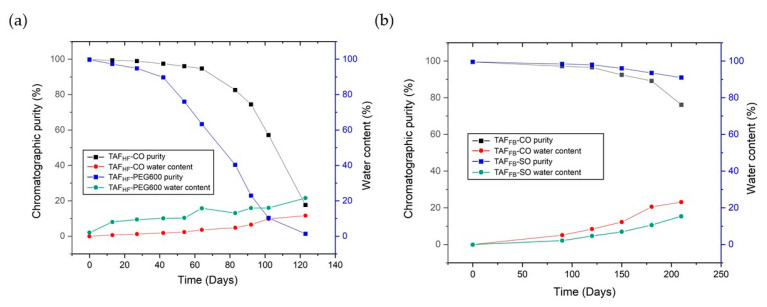
TAF purity and water uptake for the TAF_HF_ implants (**a**) and TAF_FB_ implants (**b**) as a function of time.

**Figure 6 pharmaceutics-12-01057-f006:**
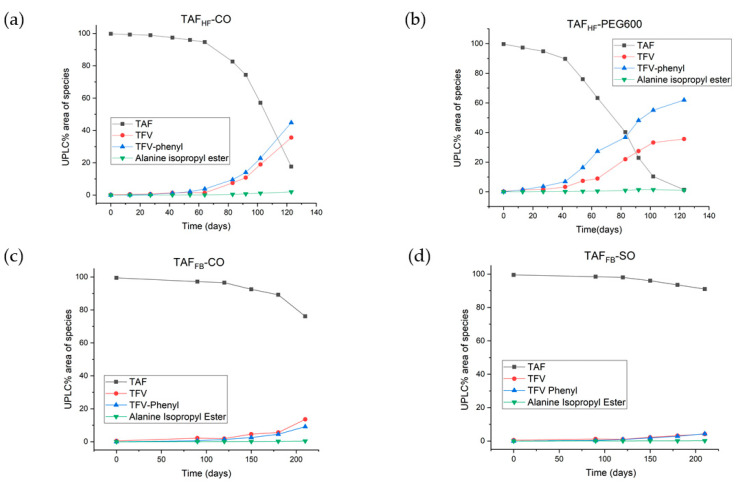
Chromatographic impurity profile of TAF_HF_-CO (**a**), TAF_HF_-PEG600 (**b**), TAF_FB_-CO (**c**), and TAF_FB_-SO (**d**) formulations.

**Figure 7 pharmaceutics-12-01057-f007:**
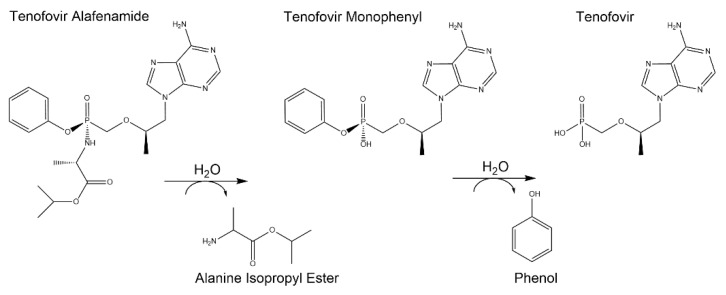
Postulated predominant TAF degradation pathway inside the implant reservoir.

**Figure 8 pharmaceutics-12-01057-f008:**
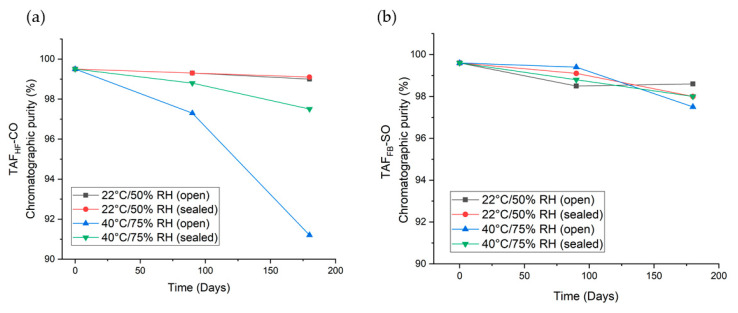
High-performance liquid chromatography (HPLC) chromatographic purity profile of 2:1 TAF_HF_-CO (**a**) and 2:1 TAF_FB_-SO (**b**) implants stored in open and closed foil pouches at 22 °C/50% RH and 40 °C/75% RH over 6-months.

**Figure 9 pharmaceutics-12-01057-f009:**
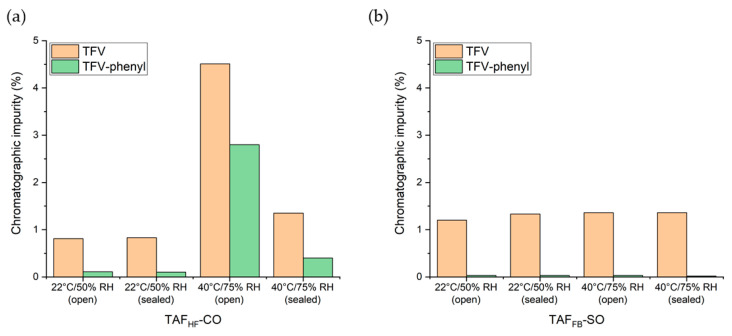
Chromatographic impurity of TAF_HF_-CO (**a**) and TAF_FB_-SO (**b**) formulations within the implant at 180 days under various storage conditions.

**Figure 10 pharmaceutics-12-01057-f010:**
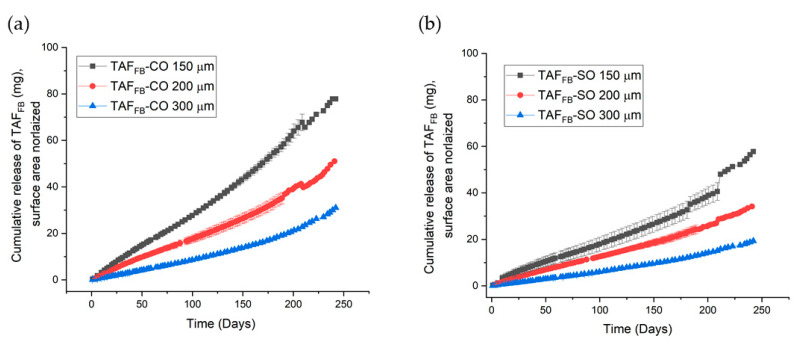
Cumulative release profiles of TAF_FB_-CO (**a**) and TAF_FB_-SO (**b**) formulations from implants of differing wall thicknesses. All samples were performed in triplicate.

**Table 1 pharmaceutics-12-01057-t001:** The solubility and chromatographic impurity of TAF_FB_ and TAF_HF_ within various pharmaceutical grade excipients identified in the FDA’s inactive ingredient list.

Excipients	TAF_HF_ Solubility (mg/mL)	TAF_HF_ % Impurities at Day-9	TAF_FB_ Solubility (mg/mL)	TAF_FB_ % Impurities at Day-9
Castor Oil	12.4	1.1	16.5	2.3
Sesame Oil	0.34	8.2	0.06	2.1
PEG600	57.6	18.5	56.8	7.2
API only	----	4.4	----	1.9
API exposed to 37 °C	----	4.4	----	2.0

**Table 2 pharmaceutics-12-01057-t002:** The formulation and PCL membrane configuration for tested implants.

Formulations	API	Excipient	Wall Thickness (µm)	Length of Implant (mm)	PCL *M*_w_ (kDa)	Approximate TAF Payload (mg)
TAF_HF_-CO	TAF_HF_	Castor oil	100	40	145	120
TAF_HF_-PEG600	TAF_HF_	PEG600				122
TAF_FB_-CO	TAF_FB_	Castor oil				118
TAF_FB_-SO	TAF_FB_	Sesame oil				113

**Table 3 pharmaceutics-12-01057-t003:** The inflection points determined by fitting the degradation profiles of various TAF formulations to the two-line models. Mass quantities of drug remaining and TAF stability at the inflection points are included.

Formulation	Inflection Point (Day)	Drug Remaining at the Inflection Point	Stability Near the Inflection Point	Water Content (mg) Near the Inflection Point	% of Water Ingress Near the Inflection Point
TAF_HF_-CO	78	91.8 mg/75.6%	93.9%	4.0	4.8%
TAF_HF_-PEG600	38	97.7 mg/79.3%	91.3%	8.4	10.1%
TAF_FB_-CO	130	82.5 mg/71.2%	96.3%	19.6	8.4%
TAF_FB_-SO	132	86.7 mg/78.7%	97.6%	10.0	4.6%

**Table 4 pharmaceutics-12-01057-t004:** The chromatographic purities of TAF_FB_-SO and TAF_FB_-CO formulations within implants of differing wall thicknesses at 210 and 240 days.

Implants	Release Rate (mg/day)	Purity at 210 Days (%)	Purity at 240 Days (%)
TAF_FB_-SO 150 µm	0.18 ± 0.03	94.3	88.8
TAF_FB_-SO 200 µm	0.11 ± 0.04	91.8	86.7
TAF_FB_-SO 300 µm	0.07 ± 0.02	96.1	93.2
TAF_FB_-CO 150 µm	0.31 ± 0.06	80.0	63.5
TAF_FB_-CO 200 µm	0.17 ± 0.07	81.2	67.9
TAF_FB_-CO 300 µm	0.10 ± 0.02	90.3	79.0

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
