# Peer review of "Performance and Stability of Tenofovir Alafenamide Formulations within Subcutaneous Biodegradable Implants for HIV Pre-Exposure Prophylaxis (PrEP)"

_pharmaceutics, 2020, doi:10.3390/pharmaceutics12111057_

Round 1
Reviewer 1 Report
Manuscript ID: pharmaceutics-975641
Title: Performance and Stability of Tenofovir Alafenamide Formulations within Subcutaneous Biodegradable Implants for HIV Pre-Exposure Prophylaxis (PrEP).
Major Comments:
- The manufacturing process of the implants involves multiple steps and a complex process.
How will this impact the cost and scalability of these implants? - There is no mention of what the total amount of drug/implant is for TAFHF and TAFFB and whether the dose/implant will lead to therapeutic efficacy for PrEP with a single implant. If a therapeutic dose is not achieved with 1 implant, how many implants will be required, and
will that be feasible clinically? - PCL has a slow degradation rate which typically takes 2-3 years depending on PCL MW and
environment. Are these implants intended to be removed after 120 days and replaced with
new implants? If yes, why not use a non-biodegradable material like EVA?
Minor Comments :
1. Line 140: It is stated that in vitro release rates were normalized to the surface area of a fullsized implant (314 mm2). Why were the rates normalized to the implant SA as opposed to the total drug loading given that the drug does not take up all the SA of the implant?
2. Supplementary Figure S1: The figure caption does not reflect the data presented for (b). The graphs compare mg TAF/day (a) vs mg TAF/day normalized to the SA but the caption says TAFHF (a) and TAFFB (b). Please correct for consistency.
3. The authors mention that TAFFB and TAFHF were formulated in a 2:1 ration with one of the 3 excipients, however the total drug loaded into each implant for these various formulations was not specified. This information needs to be added.
4. DSC analysis was performed for the polymer (PCL) but not for the drugs to assess their physical state when formulated into the implant with another excipient. In other words, are the drugs crystalline or molecularly dispersed into the implant?
5. In line 234, the authors state that “Lead excipients were identified for each API with criteria showing <3% impurity level”. First, can you clarify what is meant by “impurity level” here? Is this referring to degradation by-products? Also, TAFHF when formulated with PEG600 had impurities of 18.5% which is much greater than the <3% criterion. This formulation however was investigated in the present paper. Please clarify the rationale for including this formulation as a lead in this development.
6. In line 284, the authors state that: “As shown in Table 1, TAFHF showed higher solubility within the selected excipients than the TAFFB, resulting in higher release rates.” The data in Table 1 shows that TAFHF has a lower solubility in CO (12.4 mg/mL) and roughly the same solubility in PEG600 (57.6 mg/mL) compared to TAFFB in these same excipients (16.5 mg/mL and 56.8 mg/mL respectively). Therefore, if we compare the common excipient for TAFFB and TAFHF (i.e. CO) the solubility of TAFFB is higher but its release rates are slower than TAFHF based on the data presented in Figure 2.
7. Line 56: “compound used demonstrate the efficacy…” should be revised to “compound used to demonstrate the efficacy …”
8. Based on pH and stability data, the authors identified TAFFB-SO formulation as the lead formulation for sustained delivery of TAF; however the release rates of this formulation were the lowest amongst the four formulations investigated. Unclear if at in vitro release rates of ~0.18 mg/day these will lead to therapeutic levels in vivo for efficacy.
Author Response
We would like to thank the reviewer for the careful and thorough reading of this manuscript and for the thoughtful comments and constructive suggestions, which helped improve the quality of this manuscript. We addressed the specific queries raised in the attached document.

Reviewer 2 Report
Excellent and well described study on the TAF formulations with biodegradable implants of HIV PrEp.
Small comment: line 585 "LA-HIV" what does that mean?
Main comment: is there a possibility that also the effective anti-HIV activity of some of the collected samples can be determined? Just to know that it also remains anti-virally active in addition to the chemical detection methods? That this at least can be commented in the manuscript, or linked with a reference, as this is important.
Author Response
Reviewer 2:
“Excellent and well described study on the TAF formulations with biodegradable implants of HIV PrEp.”
The authors thank the reviewer for the kind comments and appreciate the shared enthusiasm for our work.
“Small comment: line 585 "LA-HIV" what does that mean?”
The authors changed the “LA-HIV PrEP” to “long acting HIV PrEP” on Page 16, line 603 in the revised manuscript.
“Main comment: is there a possibility that also the effective anti-HIV activity of some of the collected samples can be determined? Just to know that it also remains anti-virally active in addition to the chemical detection methods? That this at least can be commented in the manuscript, or linked with a reference, as this is important.”
We thank the reviewer for raising this important question. We didn’t directly study the anti-HIV activity of TAF within the implant in this scope. We agree that investigating the anti-HIV activity of the implants will be helpful. We have a forthcoming paper that will present the efficacy of the TAF implant in the nonhuman primate model, using a SHIV vaginal challenge.
Reviewer 3 Report
It is a well-attempted manuscript that evaluated the polycaprolactone based implants for extended delivery of Tenofovir. It is well-written in terms of scientific content.
Here are some minor comments.
- The conclusion part is very big, maybe some of the contents can be moved to the discussion part.
- If gamma irradiation was considered as an option for terminal sterilization: Have the authors tested the MW of the polymer post-irradiation? How much they have noticed. Is the drug stable to gamma irradiation, was there any degradation of the drug post-gamma irradiation? Any comments on this would be helpful to the readers.
Author Response
Reviewer 3:
“It is a well-attempted manuscript that evaluated the polycaprolactone based implants for extended delivery of Tenofovir. It is well-written in terms of scientific content.”
We thank Reviewer 3 for their positive assessment of our work and for insightful suggestions.
Here are some minor comments.
- “The conclusion part is very big, maybe some of the contents can be moved to the discussion part.”
The authors agree with the reviewer and the first two paragraphs of the conclusion have been incorporated in the discussion part. Please see Page 5 line 229, Page 7 line 283-297, Page 8, line 320, Page 11 line 444-445, Page 12 line 446-448, Page 13 line 476-487, and Page 15.
- “If gamma irradiation was considered as an option for terminal sterilization: Have the authors tested the MW of the polymer post-irradiation? How much they have noticed. Is the drug stable to gamma irradiation, was there any degradation of the drug post-gamma irradiation? Any comments on this would be helpful to the readers.”
We have tested the MW of the polymer post-irradiation. The MW data measured by Gel permeation chromatography (GPC) are shown below. The MW of the PCL showed a slight decrease in the MW after gamma-irradiation, as expected, but the sterilization process minimally affected the release rates of the implant (Johnson et al. Pharmaceutics 2019).
Type of PCL |
Sigma-PCL |
Corbion PC17 |
|
PCL extruded tubes before gamma irradiation |
Mn (kDa) |
98 |
43 |
Mw (kDa) |
145 |
107 |
|
PCL extruded tubes* post gamma irradiation |
Mn (kDa) |
70 |
35 |
Mw (kDa) |
119 |
93 |
We also investigated the purity of TAFHF and TAFFB formulations before and post-gamma irradiation. As shown in the table below, gamma irradiation does not affect the purity of the drug formulations.
Formulation |
Pre Gamma purity of TAF |
Post Gamma purity of TAF |
TAFHF-CO |
99.5% |
99.4% |
TAFHF-PEG600 |
99.4% |
99.7% |
TAFFB-CO |
99.4% |
99.2% |
TAFFB-SO |
99.4% |
98.5% |